# Phosphorylation-Mediated Activation of β-Catenin-TCF4-CEGRs/ALCDs Pathway Is an Essential Event in Development of Aggressive Hepatoblastoma

**DOI:** 10.3390/cancers14246062

**Published:** 2022-12-09

**Authors:** Ruhi Gulati, Margaret A. Hanlon, Maggie Lutz, Tyler Quitmeyer, James Geller, Gregory Tiao, Lubov Timchenko, Nikolai Timchenko

**Affiliations:** 1Division of General and Thoracic Surgery, Cincinnati Children’s Hospital Medical Center, Cincinnati, OH 45229, USA; 2Department of Neurology, Cincinnati Children’s Hospital Medical Center, Cincinnati, OH 45229, USA; 3Department of Oncology, Cincinnati Children’s Hospital Medical Center, Cincinnati, OH 45229, USA; 4Department of Surgery, University of Cincinnati College of Medicine, Cincinnati, OH 45267, USA

**Keywords:** hepatoblastoma, liver cancer, epigenetics, β-catenin

## Abstract

**Simple Summary:**

Pediatric liver cancer hepatoblastoma has a low rate of genetic mutations, suggesting additional mechanisms that involve epigenetics and signal transduction pathways. In this paper, we present evidence showing that the phosphorylation-dependent activation of wildtype β-catenin leads to the opening of genomic regions (called CEGRs/ALCDs) of oncogenes in HBL patients and to the subsequent development of liver cancer. The main activities of the ph-S675-β-catenin-CEGR/ALCD pathway are associated with the promotion of mitosis and proliferation of cancer cells. This study provides mutation-independent mechanisms that are involved in the development of aggressive HBL.

**Abstract:**

Background and Aims: Hepatoblastoma (HBL), a deadly malignancy in children, is the most common type of pediatric liver cancer. We recently demonstrated that β-catenin, phosphorylated at S675 (ph-S675-β-catenin), causes pathological alterations in fibrolamellar hepatocellular carcinoma (FLC), by activating oncogenes and fibrotic genes via human genomic regions, known as cancer-enhancing genomic regions or aggressive liver cancer domains (CEGRs/ALCDs). The aim of this study was to determine the role of the ph-S675-β-catenin-TCF4-CEGRs/ALCDs pathway in HBL. Methods: The ph-S675-β-catenin-TCF4-CEGRs/ALCDs pathway was examined in a large cohort of HBL specimens, in HBL cell lines HepG2 and Huh6, and in patient-derived xenografts (PDXs). Results: β-catenin is phosphorylated at S675 in a large portion of tested HBL patients. In these patients, ph-S675-β-catenin forms complexes with TCF4 and opens CEGRs/ALCDs-dependent oncogenes for transcription, leading to a massive overexpression of the oncogenes. The inhibition of the β-catenin-TCF4-CEGRs/ALCDs axis inhibits the proliferation of cancer cells and tumor growth in HBL cell lines and HBL-PDXs. The ph-S675-β-catenin is abundant in mitotic cells. We found that markers of HBL Glypican 3 (*GPC3*) and Alpha Fetoprotein (*AFP*) are increased in HBL patients by β-catenin-TCF4-p300 complexes. Conclusions: The phosphorylation-mediated activation of the β-catenin-TCF4-p300-CEGRs/ALCDs pathway increases oncogene expression in patients with aggressive liver cancer and promotes the development of hepatoblastoma.

## 1. Introduction

Hepatoblastoma (HBL) is the most common type of liver cancer seen in children [1,2]. A substantial number of HBL patients are faced with aggressive tumors that are characterized by metastases, chemo-resistance, and relapse [2,3]. Recent reports have shown that these cases usually represent HBL with carcinoma features and have a high mortality rate [4,5]. Compared to adult hepatocellular carcinoma (HCC), HBL is characterized by a low rate of genetic mutations that are mainly observed in *CTNNB1* (β-catenin) and *NRF2* genes [6,7]. In normal quiescent livers, β-catenin is mainly located in membranes, needing to be translocated to the nucleus to display oncogenic activities [8,9,10]. In the nucleus, β-catenin interacts with TCF4, and this interaction may also be regulated by post-translational modifications of β-catenin [10,11]. β-catenin displays its oncogenic activities by interactions with Yes-associated protein 1 (YAP1) [8]. The existence of mutation-independent oncogenic activities of β-catenin are supported by observations that overexpressed wildtype (WT) β-catenin causes liver cancers [9]. These findings showed that HBL pathogenesis involves mutation-independent mechanisms activating β-catenin and other oncogenes. In this regard, the early studies of WT β-catenin in mice demonstrated that β-catenin promotes liver proliferation after birth [12], partial hepatectomy, and liver injury [13]. These reports indicate that the overexpression of WT β-catenin causes cell proliferation, a hallmark of liver cancers. In addition, the mutation-independent oncogenic functions of β-catenin have recently been shown in liver injury [14] and in the development of cholangiocarcinoma [15]. 

Searching for mutation-independent mechanisms of pediatric liver cancers, we have identified human genomic regions (CEGRs/ALCDs) in many oncogenes. These CEGRs/ALCDs drive oncogene expressions in aggressive HBL [16,17]. Previous studies in HepG2 and Huh6 cells have shown that the inhibition of PARP1, an activator of CEGRs/ALCDs, reduced cell proliferation [17]. Our previous PDX studies revealed that the inhibition of PARP1, the subsequent suppression of CEGRs/ALCDs, and tumor growth were mediated by the reduced expression of CEGRs/ALCDs-dependent genes [18]. It has been shown that the CEGRs/ALCDs-dependent elevation of HDAC1 in HBL patients leads to epigenetic repression of hepatocyte markers, an elevation of cell proliferation markers, and a reduction of cell proliferation inhibitors such as p21 [19]. Studies of molecular mechanisms in fibrolamellar hepatocellular carcinoma (FLC) found that the FLC-specific fusion DNAJB1-PKAc protein kinase phosphorylates β-catenin at S675, resulting in an elevation of ph-S675-β-catenin-TCF4 complexes, oncogene activation, and activation of fibrotic genes via CEGRs/ALCDs [20]. The examination of GPC3 and AFP in blood samples is currently used for the diagnosis of patients with liver cancers [10,21]. Our research of β-catenin pathways in HBL patients revealed that the β-catenin-TCF4-CEGRs/ALCDs pathway accelerates the proliferation of cancer cells and increases the expression of many oncogenes, including AFP and GPC3. 

## 2. Materials and Methods

### 2.1. Pediatric HBL Patients

This study was approved by the Institutional Review Board (IRB) at CCHMC. Informed consent was obtained from each study patient (or parent as indicated) prior to obtaining specimens. Aggressive HBLs were defined as displaying one or more of the following characteristics: chemo-resistance, relapse, multiple nodules at diagnosis, vascular invasion, and lung metastases at diagnosis or after the primary surgery. In this study, we used specimens from 32 HBL patients. For Western blot, Co-Ips, and ChIP assays, we used specimens from five HBL patients who did not have mutations in CTNNB1 genes and specimens from three HBL patients who had a deletion of exon 3 in the CTNNB1 gene (Appendix A). The amounts and quality of the tissues from these patients were sufficient for such analyses. 

### 2.2. List of Antibodies Used in This Study

β-catenin (Abcam; (E247) ab32572); ph-S675-β-catenin (Cell Signaling Tech; #4176S), ph-Ser33/37/Thr41-β-catenin (Cell Signaling Tech; #9561), TCF4 (Cell Signaling, #C48H11), p300 (Invitrogen; sc: PAI-848), HDAC1 (EDM Millipore Corp. 2E10), DNAJB1 (Abcam; ab69402), p21 (Santa Cruz; sc: F5), PGAP1 (Invitrogen; sc-PA5 72340), RUNDC1 (Abcam; sc-AB 151583), HACE1 (Abcam; sc-133637), cyclin D1 (Santa Cruz; sc-753), cdc2 (Santa Cruz; sc-954), β-actin (Sigma; A5316), polyclonal GPC3 (Abcam, ab18687), monoclonal GPC3 (Abcam, ab216606), AFP (Santa Cruz, C3), cdk2 (Transduction Laboratory, C18520), and cdk4 (Abcam, ab7955). 

### 2.3. Immunohistochemistry

Liver and tumor sections from HBL patients were fixed in 4% paraformaldehyde (PFA), embedded in paraffin, and sectioned at 7 microns. The slides were deparaffinized and rehydrated through an ethanol gradient to water. Antigen retrieval was performed at high pressure and high temperature in Tris-EDTA pH 9 buffer. The samples were blocked for 1 h at room temperature. The samples were then incubated overnight at 4 °C with primary antibody diluted in blocking buffer. The next day, the samples were washed and incubated with secondary antibody diluted in blocking buffer for 4 h at room temperature, counterstained with DAPI, and mounted with FluoroMount-G (Southern Biotech 0100-01). Cultured Huh6 and HepG2 cells were fixed in 4% paraformaldehyde and IF analysis was processed as described in our previous work [16,17,18]. Briefly, the cells were blocked at room temperature for 1 h and incubated with the primary antibody overnight at 4 °C. The next day, the cells were washed and incubated with the secondary antibody at room temperature for 30 min. The nuclei were counterstained with DAPI. Images were taken using a Nikon Eclipse Ti Microscope.

### 2.4. List of TaqMan Probes Used in This Study

*CTNNB1* (Hs00355045_m1) *HACE1* (Hs00410879_m1), *NFE2L2* (Hs00975961_g1), *TNFRSF19* (Hs00218634_m1), *HMGA2* (Hs00171569_m1), *THY1* (Hs00264235_s1), *AFP* (Hs00173490_m1), *TCF7L2* (Hs01009044_m1), *REG3A* (Hs01055563_gH), *GPC3* (Hs01018937_m1) *CDK2* (Hs01548894_m1), *CDK4* (Hs01565683_g1), *MKI67* (Hs00606991_m1), *CDKN1A* (Hs00355782_m1), *RB1* (Hs01078066_m1), *TP53* (Hs01034249_m1), and *18S* (Hs03003631_g1).

### 2.5. Real-Time Quantitative Reverse-Transcriptase PCR

Total RNA was isolated from human liver samples and cell lines, cDNA was synthesized, and QRT-PCR was performed as previously described [16,17,18]. Briefly, the RNA was isolated using TRIzol/chloroform extraction. cDNA was synthesized from 2μg of RNA using the High-Capacity RNA-to-cDNA kit (Applied Biosystems™ 4387406) and diluted 5×. QRT-PCR was performed using TaqMan™ probes (see above) and TaqMan™ Gene Expression Master Mix (Applied Biosystems™ 4369016) and analyzed using the ΔΔCT method.

### 2.6. Protein Isolation, Western Blotting, Co-Immunoprecipitation

Whole cell extracts (WCEs) and nuclear extracts (NEs) were prepared as described in our previous papers [16,18]. NEs: tissues were homogenized in buffer A (10 mM Tris-HCl pH 7.5, 50 mM NaCl, 5 mM MgCl_2_, 5% β-mercaptoethanol, 10% glycerol) and centrifuged at 10,000 RPM for 10 min. The supernatant (cytoplasmic fraction) was saved, and the pellet was resuspended in buffer B (buffer A plus 0.42 M NaCl and 25% sucrose) and incubated on ice for 20 min. The samples were then centrifuged at 10,000 RPM for 10 min, and the supernatant was collected and used immediately for Western blotting or co-immunoprecipitation (Co-IP). WCEs: tissues were homogenized in buffer B and isolated in the same manner as NEs. For Western blotting, samples were loaded onto a 4–20% gradient SDS-PAGE gel, transferred to a nitrocellulose membrane, and incubated with primary and secondary antibodies. The Appendix A show whole gel images of the Western blots. Co-IP was performed using an improved True Blot protocol as previously described [17,18]. After following the normal True Blot protocol according to manufacturer’s instructions, the IP samples were boiled in loading buffer for 30 min to destroy the IgGs added during the IP protocol, reducing non-specific signals. The samples were then analyzed using the same Western blot protocol as above.

### 2.7. Pull-Down Assay

A biotinylated DNA oligomer containing the TCF4 binding site from CEGRs/ALCDs was linked to streptavidin beads and incubated with protein extracts isolated from adjacent (background) regions and tumor regions of HBL for 4 h in a buffer containing 10 mM Tris-HCL, pH 7.5, 100 mM NaCl, 5 mM β-mercaptoethanol, and 10% glycerol. The beads were washed three times with the buffer. Proteins were eluted by 1% SDS in a sample buffer and examined by Western blot with antibodies to β-catenin, TCF4, and p300, as described above.

### 2.8. Chromatin Immunoprecipitation (ChIP) Assay

Chromatin solutions were isolated from the tissues and from cultured cells. The ChIP assay was performed as described previously [17,18,19]. Briefly, after the enzymatic shearing of DNA, aliquots of chromatin solutions were incubated with antibodies to β-catenin-TCF4-p300 complexes overnight. The next day, magnetic beads were added to the samples and incubated for 2 h, and then washed to remove RNA and unbound proteins. PCR was then performed using gene-specific primers and analyzed on a 6% PAGE gel [20].

### 2.9. Proliferation Assay

The MTT proliferation assay using HepG2 and Huh6 cells was performed as described in our papers [18,19]. Briefly, HepG2 or Huh6 cells were seeded at 5 × 10^3^ cells per well in a 96-well plate. The next morning, the cells were serum-starved for 4 h using DMEM media without FBS. After starvation, fresh DMEM supplemented with 10% FBS was applied and spiked with either DMSO or 5 μM PRI-724 for 48 or 72 h. Then, fresh media was applied along with MTT and incubated for 4 h. Finally, an SDS–HCl solution was added and allowed to incubate for 18 h, and absorbance was read at 540 nm. The percent viability was calculated as compared to the DMSO control.

### 2.10. Locations of CEGRs/ALCDs within Corresponding Genes

For gene-specific locations, CEGRs/ALCDs sequences were compared against the *Homo sapiens* GRCh 38 genome build using NCBI nucleotide BLAST. 

### 2.11. Statistical Analysis

Values are presented as means ± standard deviation. A two-tailed *T*-test or one way ANOVA was utilized where appropriate to determine the statistical significance. The *p*-value < 0.05 was considered statistically significant. All statistical analyses were performed in GraphPad Prism.

## 3. Results

### 3.1. Patients with Aggressive HBL Have Increased Levels of β-Catenin, Phosphorylated at S675

Our previous studies showed that in aggressive HBL, ph-S6-p53-PARP1 increases the expression of CEGRs/ALCDs-dependent oncogenes via opening these regions for transcription [18]. However, the inhibition of PARP1 by Olaparib (Ola) was not sufficient to completely block HBL development. The RELI algorithm program has identified additional pathways of activation of CEGRs/ALCDs in many human cancers [20]. One of these pathways was described in our recent report showing that, in FLC patients, β-catenin phosphorylated at S675 forms complexes with TCF4 and that ph-S675-β-catenin-TCF4 complexes activate CEGRs/ALCDs containing oncogenes and fibrotic genes [20]. In this study, we examined whether the ph-S675-β-catenin-CEGRs/ALCDs pathway functions in patients with aggressive HBL. 

First, we searched HBL-specific genes containing CEGRs/ALCDs regions for TCF4 binding sites, identifying a strong binding site for TCF4 in all examined (*n* = 34) HBL-specific CEGRs/ALCDs. Figure 1A shows the nucleotide sequences of TCF4 sites within CEGRs/ALCDs of three cancer genes including *CTNNB1* (β-catenin) and *TCF4* genes, which code for the components of β-catenin-TCF4 complexes, as well as *AFP,* a protein that is frequently detectable in the serum of HBL patients at high levels. The presence of CEGRs/ALCDs in β-catenin and *TCF4* genes suggested that there is an auto-activation of these genes in HBL [20]. The sequences and locations of the TCF4 binding sites in 31 additional genes containing CEGRs/ALCDs are shown in Appendix A. Given the observations that ph-S675-β-catenin is a prominent player in FLC [20], we examined β-catenin expression in a fresh biobank of pediatric liver cancer samples collected at CCHMC over the last three years. QRT-PCR showed that 27 out of 30 patients have elevated levels of β-catenin mRNA (Figure 1B). Consistent with these results, Western blot analysis showed an increase in the total levels of β-catenin protein (Figure 1C). Ph-S675-β-catenin was also increased in HBL tumors. Given the critical role of GSK3β in the regulation of the stability of β-catenin by phosphorylation on S33/S37/T41, we examined whether β-catenin isoforms, phosphorylated by GSK3β, are increased in HBL.

No detectable phosphorylation of β-catenin on S33/S37/T41 residues was found in either the background or tumor sections (Figure 1C, Appendix A). The background and tumor sections from HBL patients were further analyzed by immunofluorescent (IF) staining with antibodies to ph-S675-β-catenin. This analysis found that signals of ph-S675-β-catenin in background regions were much weaker when compared to the tumor sections (Figure 1D). Images with higher magnification (20×) showed that ph-S675-β-catenin is localized in the nuclei (Figure 1D); however, it was also observed in cytoplasm (Appendix A). Thus, β-catenin mRNA and protein are elevated in the majority of the examined HBL patients, and the elevated β-catenin in HBL is phosphorylated at Ser675. To further investigate the role of ph-S675-β-catenin in hepatoblastoma, the HBL cell lines HepG2 and Huh6 were examined by IF staining for ph-S675-β-catenin expression (Figure 1E). We found that a large portion of Huh6 cells have strong ph-S675-β-catenin signals located mainly in the cellular membranes (Figure 1E, left; Appendix A). Interestingly, the ph-S675-β-catenin signals were increased mainly in mitotic Huh6 cells (Figure 1E). No mitotic Huh6 cells negative for ph-S675-β-catenin were observed. The intensive staining of ph-S675-β-catenin was also found in mitotic HepG2 cells (Figure 1E, right). The ph-S675-β-catenin was located in the nuclei and cytoplasm of the HepG2 cells (Appendix A). 

### 3.2. β-Catenin-TCF4-p300 Complexes Increase the Expression of CEGRs/ALCDs-Containing Oncogenes in Patients with Aggressive HBL 

Since ph-S675-β-catenin-TCF4-p300 complex activated CEGR/ALCDs-containing genes in FLC [20], we examined whether these complexes might activate CEGR/ALCD-containing oncogenes in aggressive HBL. First, the expression of β-catenin and TCF4 was analyzed in freshly resected aggressive HBL samples (*n* = 5), including three samples lacking deletion/mutation in exon 3 of β-catenin (Appendix A). Since histone acetylase p300 opens chromatin by the acetylation of histone H3 on K9, we also examined the expression of p300 in these samples. A Western blot analysis confirmed that WT β-catenin is elevated in tumor sections of HBL patients (Figure 2A, Appendix A). TCF4 was also increased in HBL tumors compared to background regions, while p300 expression was not altered. Co-IP studies showed that p300, TCF4, and β-catenin form a complex in aggressive HBL tumors (Figure 2A, bottom, Appendix A). To determine whether β-catenin-TCF4-p300 complex binds to the CEGRs/ALCDs, a pull-down assay with a DNA oligomer containing a TCF4-binding site was performed (Figure 2B, Appendix A). These studies revealed that β-catenin-TCF4-p300 complex binds to the CEGRs/ALCDs in vitro. To examine whether this interaction occurred in HBL patients, we applied the chromatin immunoprecipitation (ChIP) approach and found that the β-catenin-TCF4-p300 complex is bound to CEGRs/ALCDs in five examined genes (Figure 2C). Using our fresh biobank of pediatric liver cancer specimens (*n* = 32), we examined whether mRNAs of CEGRs/ALCDs encoded genes are elevated in the liver tumors of HBL patients. As shown in Figure 2D,E, mRNAs encoded by the genes containing CEGRs/ALCDs, *HACE1*, *NRF2*, *TNFRSF19*, *TCF4, Thy1, HMGA2*, and *AFP* were elevated in a large portion of samples from the fresh biobank. The *TNFRSF19 and HMGA2* genes were included in these studies since it has been recently shown that *TNFRSF19* is a marker of HBL [22] and *HMGA2* has also been shown as potential marker of HBL [23]. Both genes contain CEGRs/ALCDs (Appendix A). The mRNAs encoding *TNFRSF19*, *HMGA2*, *CTNNB1, Thy1*, and *AFP* are elevated in the majority of the examined patients (Figure 1B and Figure 2E). QRT-PCR showed that tumor sections of HBL patients also often have increased *TCF4*. Among the elevated CEGRs/ALCDs-containing genes, we identified the *AFP* gene. The AFP data from pathology reports revealed that HBL patients with an active β-catenin-TCF4-CEGRs/ALCDs pathway had high levels of AFP protein in their serum samples, consistent with other reports that increased AFP is a signature of liver cancer. We also asked whether known targets of β-catenin that do not have CEGRs/ALCDs might be increased in aggressive HBL with activated ph-S675-β-catenin-TCF4 pathway. Previous studies showed that REG3A is elevated in HCC and HBL patients and that this elevation correlated with the activation of β-catenin [24]. We found that the *REG3A* gene does not have CEGR/ALCD, but the promoter of *REG3A* gene contains two strong TCF4 consensuses near the start of transcription (Appendix A). QRT-PCR analysis showed that REG3A mRNA is highly elevated in 19 out of 21 patients with activated ph-S675-β-catenin-TCF4 pathway (Figure 2E). Thus, the examination of β-catenin-TCF4-CEGRs/ALCDs-dependent oncogenes in a large cohort of HBL patients showed that this pathway is functioning in patients with aggressive HBL, and that the downstream targets of this pathway are elevated in HBL tumors. 

### 3.3. Expression of Glypican 3, a Target of β-Catenin, Is Strongly Elevated in Patients with Active β-Catenin-TCF4-CEGRs/ALCDs Pathway

GPC3 is increased in the majority of HCC and HBL patients and, like AFP, is detectable in the serum of both patient populations [21]. Therefore, we asked whether the elevation of *GPC3* in HBL patients is associated with β-catenin-TCF4-CEGRs/ALCDs-mediated opening of the chromatin around the *GPC3* gene. The *GPC3* gene is long and is located within a 500,000 bp DNA region on chromosome X. The gene contains eight short exons, separated by seven long introns. BLAST search analyses identified six intronic CEGRs/ALCDs in the *GPC3* gene (Figure 3A). These CEGRs/ALCDs share 80–88% homology with the CEGR/ALCD in *NRF2* gene. The most critical feature of CEGRs/ALCDs in the *GPC3* gene is that all of them contain an identical, perfect β-catenin-TCF4 site (Figure 3A, blue box). To examine if β-catenin-TCF4 complexes bind to the CEGRs/ALCDs of *GPC3* gene in tumors of HBL patients, we performed a ChIP assay with primers to the CEGR/ALCD in intron 2 of the *GPC3* gene. This CEGR/ALCD was selected since sequences of other CEGRs/ALCDs of the *GPC3* gene are highly homologous, making it difficult to generate specific primers. We found that the β-catenin-TCF4-p300 complex occupies the CEGR/ALCD of *GPC3* in the tumor sections of HBL patients, but not in the background regions (Figure 3B). The analysis of *GPC3* expression in the specimens of the fresh biobank using QRT-PCR, immunostaining, and Western blot analyses showed high levels of both *GPC3* mRNA (Figure 3C) and GPC3 protein (Figure 3D,E) in the tumor sections of HBL patients compared to the background sections. We learned that the GPC3 protein is sensitive to the degradation/cleavage that occurs during the extraction of proteins from tissues. Therefore, we established a protocol that prevents the degradation of GPC3 during protein isolation (methods in Appendix A). The examination of GPC3 in 12 HBL patients using this protocol revealed that full-length GPC3 is elevated in all examined tumors. Figure 3F (Appendix A) shows the levels of GPC3 in six fresh HBL specimens. In some HBL patients, we observed two GPC3 isoforms with MW 70 kD and 90 kD, suggesting the possibility of post-translational modifications of GPC3 in HBL patients (Appendix A). Taken together, the activation of the β-catenin-TCF4 pathway in aggressive HBL leads to the opening of CEGRs/ALCDs in *GPC3* and to the increased expression of GPC3.

Since GPC3 is activated in HBL by ph-S675-β-catenin-TCF4-CEGRs/ALCDs pathway, we asked whether the tumors of these patients might secrete GPC3 into the bloodstream. We found that GPC3 is elevated in liver tumors and in plasma from the same HBL patient #8 (Figure 3G, Appendix A). Thus, increased GPC3 produced in the liver tumor of this patient is secreted in the bloodstream. We also examined GPC3 protein in the plasma of additional HBL patients (*n* = 7) with an active ph-S675-β-catenin-TCF4-CEGRs/ALCDs pathway. As shown in Figure 3G (Appendix A), full-length GPC3 was detected in the analyzed HBL plasma samples. Thus, GPC3 is secreted into the bloodstream from liver tumors in HBL patients with an active β-catenin-TCF4-GPC3 pathway.

### 3.4. Inhibition of β-Catenin by PRI-724 Removes β-Catenin-TCF4-p300 Complexes from CEGRs/ALCDs and Represses Expression of CEGRs/ALCDs-Dependent Oncogenes

To examine a possible causal role of ph-S675-β-catenin in the activation of CEGRs/ALCDs-dependent oncogenes, we thought to inhibit β-catenin activity in HepG2 and Huh6 cell lines by PRI-724 and examine the effect of this inhibition on the β-catenin-TCF4-p300-CEGRs/ALCDs pathway. PRI-724 was selected for these studies since it is currently being used in clinical trials for patients with liver fibrosis/cirrhosis [25]. We first examined the dose-dependent effects of PRI-724 on the proliferation of HepG2 and Huh6 cells and found that 5μM PRI-724 is optimal for the inhibition of the proliferation in these cells [20]. Under these conditions, Co-IP studies displayed that PRI-724 destroys ph-S675-β-catenin-TCF4-p300 complexes (Figure 4A, Appendix A). 

We next asked if the expression of β-catenin-TCF4-CEGRs/ALCDs-dependent genes is reduced in cells with inhibited β-catenin activity. Western blotting revealed that the protein levels of the CEGRs/ALCDs-containing genes HACE1, HDAC1, DNAJB1, p53, PGAP1, and NRF2 are all reduced in cells treated with PRI-724 (Figure 4B, Appendix A). Examination of the status of CEGRs/ALCDs in six oncogenes showed that the destruction of β-catenin-TCF4-p300 complexes by PRI-724 leads to the removal of these complexes from the CEGRs/ALCDs, and that the repression of these DNA regions includes the CEGR/ALCD of the *p53* gene (Figure 4C,D). Note that in cancer cells, the *p53* gene produces the oncogenic ph-S6-p53 form [20]. Figure 4E summarizes our studies of CEGRs/ALCDs-dependent genes in cultured cells generated from HBL patients.

### 3.5. PRI-724-Mediated Inhibition of β-Catenin-TCF4-CEGRs/ALCDs axis Reduces Proliferation of Cancer Cells

To investigate the consequences of ph-S675-β-catenin inhibition, we examined the proliferation of Huh6 and HepG2 cells treated with PRI-724. For this goal, the cells were treated with PRI-724 for 48 or 72 h and the proliferation of these cells was examined by several approaches. First, we determined the mRNA and protein levels of the proliferation markers. Figure 5A shows that the mRNA levels of *CDK2*, *CDK4,* and *Ki67* are reduced in cells with inhibited β-catenin, while the mRNA and protein levels of the inhibitor of the proliferation of *CDKN1A (p21)* are elevated. We also detected the reduction of *Rb1* and *Tp53* mRNAs in HepG2 cells. Second, Western blotting confirmed the reduction of cyclin D1, cdk2, cdk4, and cdc2 as well as the elevation of p21 and HNF4 proteins in cells with inhibited β-catenin activity (Figure 5B, Appendix A). Third, an MTT cell proliferation assay showed that PRI-724 causes the inhibition of the proliferation of both HepG2 and Huh6 cells by approximately 50–60% (Figure 5C). Fourth, the bright-light images of DMSO-treated cells and cells treated with PRI-724 for 72 h after the initiation of treatments reveal that DMSO-treated Huh6 and HepG2 cells proliferate and form large cell clusters, while PRI-724 treatments inhibit cell proliferation and reduced cell cluster size (Figure 5D). Since ph-S675-β-catenin is mainly observed in mitotic HepG2 and Huh6 cells (Figure 1E), we examined whether PRI-724 affects mitotic cells using IF analysis. In addition, a target of the ph-S675-β-catenin-CEGR/ALCDs pathway, GPC3, was examined in these studies. DMSO-treated Huh6 cells were found to have 8–10 mitotic figures per colony, while HepG2 cells had 2–3 mitotic figures per colony. Both were positive for ph-S675-β-catenin and GPC3 (Figure 5E,F), while cells with inhibited β-catenin had no mitotic figures. In rare cases, PRI-724-treated Huh6 cells had one mitotic figure per cluster. The calculations of a percentage of ph-S675-β-catenin positive cells with mitotic figures showed a dramatic reduction of mitosis/proliferation in cells with inhibited β-catenin (Figure 5G). Thus, PRI-724 reduces the amounts of ph-S675-β-catenin-positive mitotic cells, indicating the inhibition of the proliferation of cancer cells.

In agreement, a target of this pathway, GPC3, was reduced in PRI-724 treated cells (Figure 5E,F). In summary, several independent approaches revealed that the inhibition of the β-catenin-TCF4-CEGRs/ALCDs pathway reduces the proliferation of hepatoblastoma cells. 

### 3.6. Inhibition of β-Catenin in Hepatoblastoma Cells Removes β-Catenin-TCF4-p300 Complex from CEGR/ALCD of GPC3 and from the Promoter of AFP Gene and Represses Expression of GPC3 and AFP

Given the strong correlations of GPC3 and AFP elevation with the activation of the β-catenin-TCF4-CEGRs/ALCDs pathway in HBL patients and the identification of GPC3-positive mitotic figures in cultured cells (Figure 5), we performed detailed molecular studies of this pathway. The HepG2 and Huh6 cells were treated with 5μM PRI-724, and the AFP and GPC3 mRNA and protein levels were examined. QRT-PCR showed that the inhibition of β-catenin reduced *AFP* and *GPC3* mRNA levels (Figure 6A). Western blotting showed that the GPC3 and AFP protein levels were reduced in the treated HepG2 and Huh6 cells (Figure 6B, Appendix A). The immunostaining of HepG2 and Huh6 cells showed a reduced GPC3 signal in cells treated with PRI-724 (Figure 6C). We next asked if the inhibition of β-catenin removes β-catenin-TCF4-p300 complexes from the CEGRs/ALCDs of the *GPC3* gene and from the *AFP* promoter using a ChIP assay. Although the *AFP* gene contains a CEGR/ALCD in intron 3, we found that the promoter of the *AFP* gene contains a perfect TCF4 binding site close to the start of transcription (−281 bp, Figure 7B). The ChIP assay demonstrated that in both HepG2 and Huh6 cells, the PRI-724 treatment removed β-catenin-TCF4-p300 complexes from the CEGR/ALCD in *GPC3* and from the promoter of *AFP* (Figure 6D,E). Figure 6F summarizes the results of these studies. We found that the inhibition of β-catenin in cancer cells leads to the dissociation and removal of β-catenin-TCF4-p300 complexes from the CEGRs/ALCDs in the *GPC3* gene and from the *AFP* promoter, resulting in the suppression of these genes and subsequent reduction of AFP and GPC3 mRNAs and proteins.

### 3.7. Silencing CEGRs/ALCDs in HBL-Derived PDXs Inhibits Expression of GPC3 and AFP in Tumors

CEGRs/ALCDs genomic regions contain binding sites for different transcription factors [17,18], including those that have CEGRs/ALCDs in their own genes such as β-catenin and TCF4 (Figure 1A). Therefore, the regulation of the CEGRs/ALCDs-dependent genes might involve communications between different transcription factors and signaling pathways. For example, ph-S6-p53-PARP1 complexes activate the CEGRs/ALCDs-containing genes in the PDXs [18]. We have previously generated PDXs using HBL tumor samples with elevated ph-S6-p53-PARP1 complexes. These PDXs were treated with an inhibitor of PARP1 Olaparib (Ola) to reduce the activities of ph-S6-p53-PARP1 complexes. We found that Ola inhibits tumor growth by removing ph-S6-p53-PARP1 complexes from many CEGRs/ALCDs-containing genes including β-catenin (Figure 1A) [18]. To determine whether Ola reduces the targets of the ph-S675-β-catenin-CEGRs/ALCDs pathway in PDXs, we examined the GPC3 and AFP levels in the tumor tissues of PDXs (*n* = 6) from three HBL patients treated with vehicle (control) or with Ola. The total β-catenin and ph-S675-β-catenin levels were also measured in the same samples. Western blotting showed that β-catenin, ph-S675-β-catenin, GPC3, and AFP were reduced in Ola-treated PDXs compared to vehicle-treated PDXs (Figure 7A, Appendix A). To examine whether the reduction of β-catenin, AFP, and GPC3 is associated with the repression of CEGRs/ALCDs, we performed a ChIP analysis.

The sequence analysis of the *AFP* gene showed that, in addition to CEGR/ALCD in the intron 3, the promoter of this gene contains four TCF4 binding sites (Figure 7B). For the ChIP assay with the *AFP* gene, we used primers covering the TCF4 binding site 1 that was in close proximity to the start of transcription. The examination of ph-S6-p53-PARP1 complexes on the β-catenin CEGR/ALCD showed that these complexes are removed from the CEGR/ALCD of β-catenin in the Ola-treated PDXs (Figure 7C). This leads to the repression of the β-catenin gene, since H3K9 in this region is methylated. The examination of the complexes on CEGR/ALCDs of the *GPC3* gene and on the *AFP* promoter showed that the Ola-mediated inhibition of β-catenin resulted in the removal of β-catenin-TCF4-p300 complexes from CEGR/ALCD of the *GPC3* gene and from the AFP promoter and the consequent repression of these genes. Taken together, these results suggest that the Ola-mediated inhibition of CEGRs/ALCDs in HBL-PDXs reduces the network of CEGRs/ALCDs-containing cancer-associated genes. In particular, we found that Ola removes ph-S6-p53-PARP1 complex from the CEGR/ALCD of the β-catenin gene, leading to β-catenin reduction. The inhibition of β-catenin, in turn, leads to the reduction of β-catenin-TCF4-p300 complexes and the repression of oncogenes including *GPC3* and *AFP* genes (Figure 7E).

## 4. Discussion

Hepatoblastoma, the most common pediatric liver cancer, is extensively studied, but effective therapeutic approaches for aggressive HBL cases are not available. Since β-catenin is frequently mutated in HBL tumors, previous studies have focused on examining the β-catenin mutations in HBL. These studies have provided a solid foundation for researchers regarding β-catenin; however, it is become clear from recent studies that the development of HBL includes mechanisms independent of β-catenin mutations. Our lab has conducted detailed mechanistic studies of HBL, focusing on the post-translational modifications of oncogenes. We have found that the CEGRs/ALCDs regions of the human genome are key determinants of aggressive HBL. These regions are short, highly homological DNA sequences that have several 100% homologous subregions containing binding sites for various transcriptional complexes. In this paper, we present evidence that the β-catenin-TCF4 complex increases the expression of CEGRs/ALCDs-dependent oncogenes in HBL patients, HBL cell lines, and PDX mouse models of HBL. Although we have concentrated our studies on a small number of CEGRs/ALCDs-containing oncogenes, this pathway may function through many CEGRs/ALCDs-containing genes because TCF4 binding sites are observed in all identified CEGRs/ALCDs (Appendix A). An analysis of fresh biobank HBL specimens revealed that over 70% of the examined HBL patients with aggressive forms of the disease have a significant increase in β-catenin mRNA and protein, which is phosphorylated at Ser675. TCF4 expression is also increased in some of these patients. In these patients, abundant β-catenin-TCF4-p300 complexes bind to CEGRs/ALCDs, allowing the transcription of CEGRs/ALCDs-containing regions. It is worth noting that previous ChIP-Seq studies on colon cancer cells found β-catenin-TCF4 on the genomic regions of HDAC1 and DNAJB1 [26]. We examined blood samples from HBL patients with active β-catenin-TCF4 pathways in primary liver cancer and discovered GPC3 and AFP in their bloodstream. Regarding additional potential biomarkers of aggressive HBL, Zhou and colleagues examined 277 with altered expression in HBL patients and discovered that TNFRSF19 is highly upregulated and could be used as an HBL biomarker [22]. TNFRSF19 contains a CEGR/ALCD in the first intron, and TNFRSF19 is highly increased in 26 of 29 HBL patients, according to our findings (Figure 2E). These findings suggest that TNFRSF19 may be another biomarker that is upregulated by β-catenin-TCF4 complexes in HBL. It should be noted that our study is focused on providing proof-of-principle for the reduction of the carcinogenic activation of CEGRs/ALCDs in aggressive HBL via β-catenin inhibition, rather than on the development of clinically effective and safe drug doses. In this regard, a recent report of a multicenter phase 1/2a trial for patients with cirrhosis found that low doses of PRI-724 significantly improved liver stiffness and reduced serum albumin levels, but not fibrosis [27]. 

The HBL-PDX models are best for reproducing the in vivo biology of HBL. Using this system, we discovered that oncogenes and other cancer-associated genes are part of a network of gene expression regulation that is dependent on CEGR/ALCDs. Genes dependent on CEGRs/ALCDs regulate both themselves and other genes via CEGRs/ALCDs. Among these are the genes p53, TCF4, and β-catenin. Our earlier research demonstrated that p53, which is phosphorylated at Ser6 by JUNK1/2 kinases, form complexes with PARP1, which promotes the expression of CEGRs/ALCDs-dependent genes in patients with aggressive HBL [18,19]. We have discovered that the ph-S6-p53-PARP1 complexes also activate β-catenin in PDXs, and the β-catenin and its downstream targets are expressed less when PARP1 is inhibited by Ola (Figure 7). The downstream targets of the β-catenin-TCF4-CEGRs/ALCDs pathway include GPC3 and AFP. The GPC3 and AFP levels in tumors are decreased as a result of Ola-mediated suppression. Given that GPC3 and AFP both enhance the proliferation of HepG2 cells via the HuR-mediated regulation of apoptosis, it is likely that the PRI-724-mediated reduction of GPC3 and AFP in HepG2 and HUH6 cells is responsible for this inhibition [28,29]. Moreover, a recent report suggested that a combined elevation of GPC3/AFP in serum may be diagnostic for HCC [30]. Although the focus of our studies was the β-catenin-CEGRs/ALCDs pathway, it is likely that the promoters of cancer-associated genes, such as the AFP gene, have TCF4 binding sites and may be activated by β-catenin without the assistance of CEGRs/ALCDs. In agreement with this suggestion, our examination of CEGRs/ALCDs-independent *REG3A* showed that its mRNA is highly elevated in HBL patients with an active ph-S675-β-catenin-TCF4 pathway (Figure 2E). 

## 5. Conclusions

In this study, we investigated five HBL patients in depth who had no β-catenin mutations and three HBL patients with deletion of exon 3 of the β-catenin gene. The exon 3 was deleted in one allele, but the patients produced WT β-catenin in the other allele. Regardless of mutation status, all eight HBL patients exhibited expressed ph-S675-β-catenin in tumor sections. Thus, we believe that the phosphorylation of β-catenin at Ser675 is one of the key events in the development of aggressive HBL. It remains to be investigated whether the mutant β-catenin is also phosphorylated at Ser675 and this phosphorylation in the mutant form also enhances its oncogenic activities. In summary, the studies of the ph-S675-β-catenin-TCF4-CEGRs/ALCDs pathway in HBL patients, in hepatoblastoma cell lines HepG2 and Huh6 and in PDX models of HBL, demonstrated the essential role of the ph-S675-β-catenin-TCF4-CEGRs/ALCDs axis in the development of aggressive HBL.

## Figures and Tables

**Figure 1 cancers-14-06062-f001:**
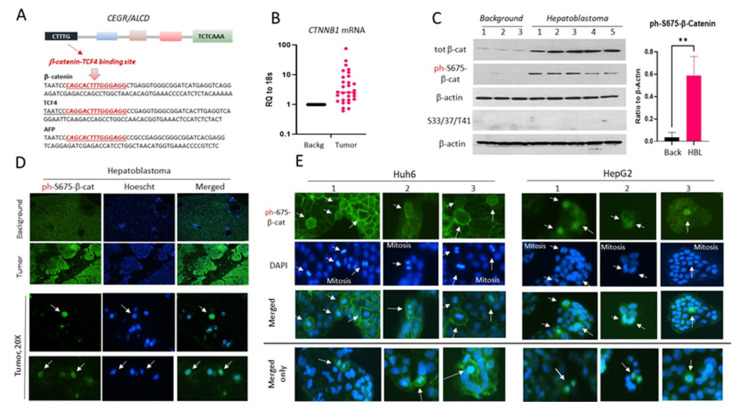
Ph-S675-β-catenin is increased in patients with aggressive HBL and in HBL cell lines HepG2 and Huh6. (**A**) Structure of the CEGRs/ALCDs in cancer genes is shown. Black box shows TCF4 binding sites in the CEGRs/ALCDs of β-catenin, *TCF4*, and *AFP* genes. (**B**) Levels of *CTNNB1* (β-catenin) mRNA in fresh HBL biobank (*n* = 32) (*p* = 0.0127, unpaired *t*-test) (**C**) Levels of total β-catenin and ph-S675-β-catenin in background and tumor regions of HBL. Bar graph shows levels of ph-S675-β-catenin as a ratio to β-actin (*p* = 0.0017, unpaired *t*-test). (** *p* ≤ 0.05) (**D**) IF staining of livers of HBL patients with antibodies to ph-S675-β-catenin and stained with DAPI. Arrows indicate nuclear localization of ph-S675-β-catenin. (**E**) ph-S675-β-catenin is elevated in mitotic (shown by arrows) Huh6 and HepG2 cells. The uncropped Western blots have been shown in Appendix A.

**Figure 2 cancers-14-06062-f002:**
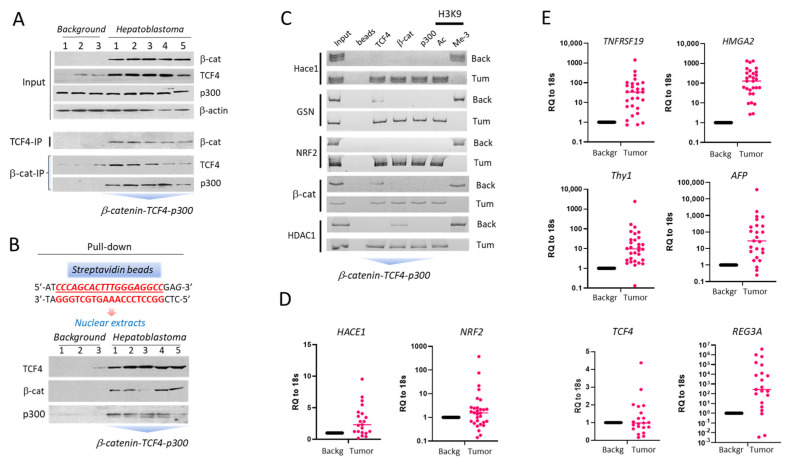
The β-catenin-TCF4 pathway is activated in patients with HBL and increases expression of CEGRs/ALCDs-containing genes. (**A**) Co-IP studies. Input: levels of β-catenin, TCF4, p300 in protein extracts used for IP. Bottom images show results of Co-IP studies. (**B**) Pull-down assay with a DNA oligomer containing a conserved TCF4 binding site identified in all studied CEGRs/ALCDs. (**C**) ChIP assay with CEGRs/ALCDs of five oncogenes. B: beads, in: input. Chromatin solutions from two HBL patients expressing only WT β-catenin were used. (**D**,**E**) QRT-PCR analyses shows mRNA levels of *HACE1* (*p* = 0.0006), *NRF2* (*p* = 0.1985), *TNFRSF19* (*p* = 0.0407), *HMGA2* (*p* = 0.0002), *Thy1* (*p* = 0.2022), *AFP* (*p* = 0.2586), *TCF4* (*p* = 0.2649), and *REG3A* (*p* = 0.1503) (unpaired *t*-tests). The uncropped Western blots have been shown in Appendix A.

**Figure 3 cancers-14-06062-f003:**
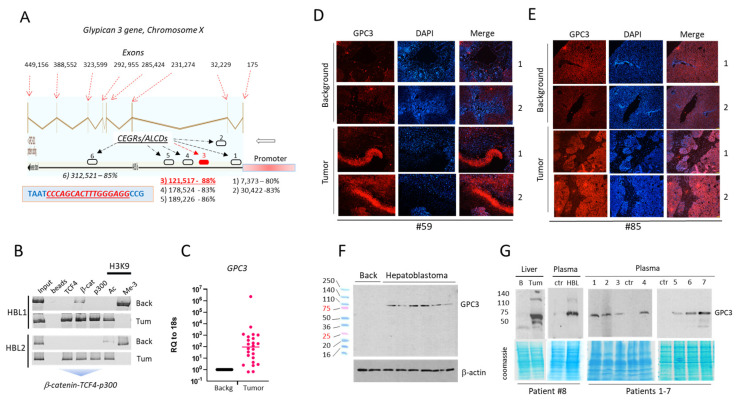
GPC3 is elevated in HBL patients by β-catenin-TCF4-p300 pathway via CEGRs/ALCDs in *GPC3* gene. (**A**) Structure of the *GPC3* gene. The 100% homologous TCF4 binding site is shown in the blue box. (**B**) ChIP assay with the third CEGR/ALCD of *GPC3*. (**C**) *GPC3* mRNA expression in a fresh biobank of HBL samples (*n* = 32) (*p* = 0.3202, unpaired *t*-test). (**D**,**E**) IF staining of pediatric liver cancer sections with antibodies to GPC3. (**F**) GPC3 expression in specimens from a fresh biobank was determined by Western blot. (**G**) Levels of GPC3 in blood of eight HBL patients with active ph-S675-β-catenin-TCF4-CEGRs/ALCDs-GPC3 pathway in liver tumors. The uncropped Western blots have been shown in Appendix A.

**Figure 4 cancers-14-06062-f004:**
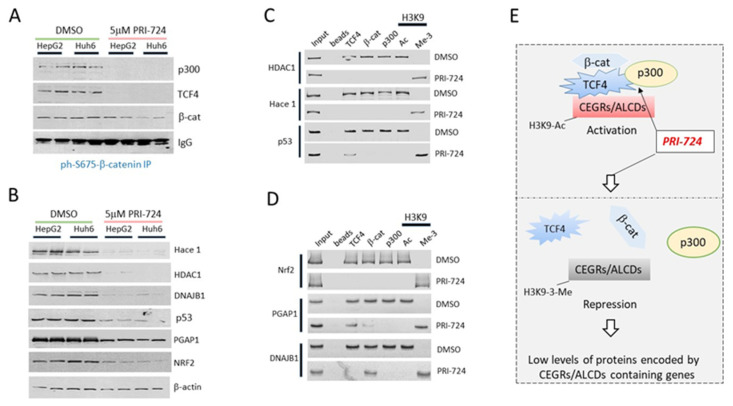
Inhibition of β-catenin in HBL cells removes β-catenin-TCF4-p300 complexes from CEGRs/ALCDs and reduces expression of CEGRs/ALCDs-containing oncogenes. (**A**) Co-IP shows that PRI-724 destroys β-catenin-TCF4-p300 complexes in HepG2 and Huh6 cells. (**B**) Expression of proteins (shown on the right) determined by Western blotting in DMSO and PRI-724-treated HepG2 and Huh6 cells. (**C**,**D**) ChIP assay with CEGRs/ALCDs of six oncogenes shown on the left. (**E**) A summary of the results of the ChIP assay. The uncropped Western blots have been shown in Appendix A.

**Figure 5 cancers-14-06062-f005:**
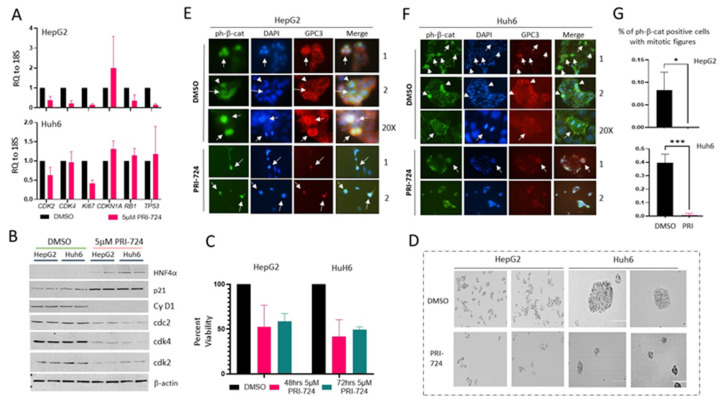
PRI-724 inhibits proliferation of Huh6 and HepG2 cells. (**A**) mRNA levels coding for cell cycle proteins by QRT-PCR. (**B**) Expression of cell cycle proteins determined by Western blot. (**C**) MTT cell proliferation assay with HepG2 (48 h, *p* = 0.0142; 72 h, *p* = 0.0258) and Huh6 (48 h, *p* = 0.0012; 72 h, *p* = 0.0025) cells treated with DMSO and PRI-724 (one-way ANOVA). (**D**) Images of cells at 72 h after addition of DMSO or PRI-724. Scale bar is 200μm. (**E**,**F**) Huh6 and HepG2 cells treated with DMSO or PRI-724 were examined by IF with antibodies to ph-S675-β-catenin and GPC3 and stained with DAPI. Arrows indicate mitotic figures. (**G**) Calculations of percentage of ph-S675-β-catenin-positive HepG2 and Huh6 with mitotic figures treated with DMSO or PRI-724. Three large (10×) fields were counted for each treated cell line (HepG2 *p* = 0.0219, Huh6 *p* = 0.0005, unpaired *t*-tests) (* *p* ≤ 0.05, *** *p* ≤ 0.001). The uncropped Western blots have been shown in Appendix A.

**Figure 6 cancers-14-06062-f006:**
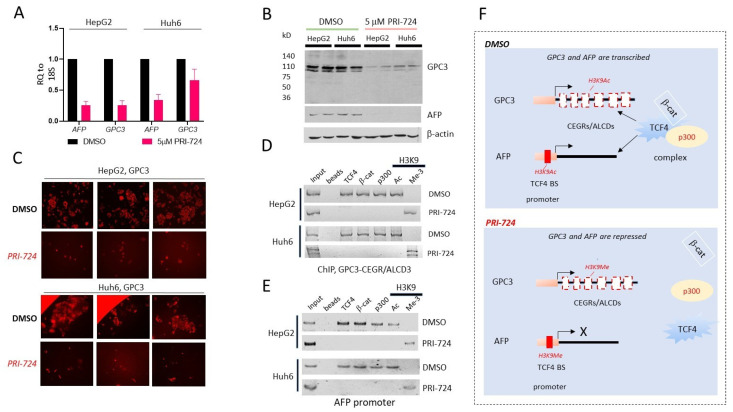
PRI-724-mediated inhibition of β-catenin activity in cancer cells reduces expression of GPC3 and AFP. (**A**) QRT-PCR of *AFP* and *GPC3* in HepG2 (*AFP*, *p* = 0.000037; *GPC3*, *p* = 0.000059) and Huh6 (*AFP*, *p* = 0.000240; *GPC3*, *p* = 0.028805) cells (unpaired *t*-tests). (**B**) Western blots of GPC3 and AFP in DMSO or PRI-724-treated HepG2 and Huh6 cells. (**C**) Immunostaining of DMSO and PRI-724-treated HepG2 and Huh6 cells with antibodies to GPC3. (**D**,**E**) ChIP assay with the third CEGR/ALCD of the *GPC3* gene (**D**), and *AFP* promoter (**E**). (**F**) Hypothetical mechanism of repression of the *AFP* and *GPC3* genes in cancer cells treated with PRI-724.

**Figure 7 cancers-14-06062-f007:**
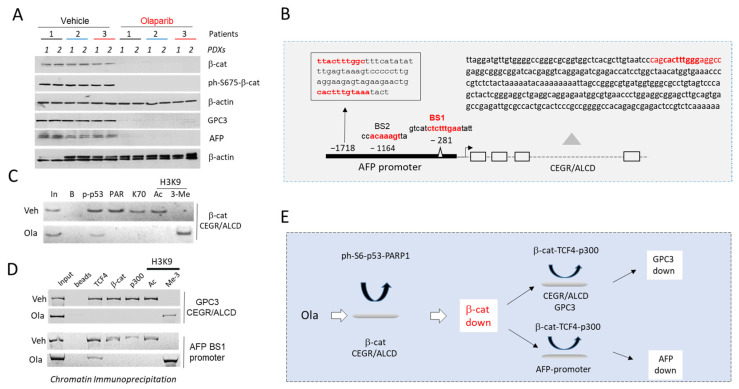
Repression of CEGRs/ALCDs in HBL-derived PDXs inhibits expression of β-catenin, GPC3, and AFP in tumors and reduces secretion of AFP in bloodstream. (**A**) Levels of β-catenin, ph-S675-β-catenin, GPC3 and AFP in tumors of PDXs treated with vehicle and Olaparib (Ola). (**B**) Locations of CEGR/ALCD and TCF4-binding sites in the *AFP* gene. (**C**) Examination of ph-S6-p53-PARP1 complexes on the CEGR/ALCD of β-catenin gene by ChIP assay. (**D**) Examination of β-catenin-TCF4-p300 complexes on CEGR/ALCD of *GPC3* gene and on AFP promoter by ChIP assay. (**E**) A diagram showing a cascade of CEGRs/ALCDs-dependent events in HBL-PDXs treated with Ola. The uncropped Western blots have been shown in Appendix A.

## Data Availability

Not applicable.

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
