# Peer review of "Phosphorylation-Mediated Activation of β-Catenin-TCF4-CEGRs/ALCDs Pathway Is an Essential Event in Development of Aggressive Hepatoblastoma"

_cancers, 2022, doi:10.3390/cancers14246062_

Round 1
Reviewer 1 Report
The authors of this publication investigated the role of beta-catenin in the development and carcinogenesis of hepatomas. Huh6 and HepG2 cell lines, among others, were utilized to examine the effects of beta-catenin in tumor development. Further, beta-catenin phosphorylation was markedly increased in all cell lines. In addition, it was shown that hepatoma cells had elevated amounts level of TCF4-CEGR and ALDCD regulatory axis. However, The results were not validated using the in vivo studies.
1-Since the role of b-catenin-CEGRs/ALCDs in the development of HCC has previously been implicated in several previous research, the originality of the current work is rather limited.
doi.org/10.1002/hep4.2055
DOI: 10.1002/hep4.2055
DOI: 10.36468/pharmaceutical-sciences.602
2-Cells from the hepatoma cell lines, HepG2 and Huh6 cell lines are frequently used as in vitro models. I strongly suggest authors include both hepatoma cells and a normal primary cell line to support the results of a study (A comparison of p-beta-catenin-TCF4-CEGRs/ALCDs expression and pathway analysis, at the very least).
3-The immunofluorescence stainings, which the authors used to evaluate the change in b-Catenin expression in Hepatoma cells, should also be quantified.
4- The quality images of immunofluorescence in Figures 3D and 6C need to be used.
5- In Figure 3 D, the authors also need to compare GPC3 expression in the normal vs tumor areas.
Author Response
We thank the reviewer for useful recommendations for improvement of the manuscript. Detailed responses to the comments are shown below.
Comment 1: Since the role of b-catenin-CEGRs/ALCDs in the development of HCC has previously been implicated in several previous research, the originality of the current work is rather limited.
Response: We agree with the reviewer that the role of b-catenin-CEGRs/ALCDs has been described for fibrolamellar HCC, but we think that our work is innovative in following aspects:
- Our MS shows that the b-catenin-ALCD pathway is involved in the pediatric liver cancer HBL.
- Our MS shows for the first time that phosphorylated (ph-S675)-b-catenin is an essential player of pediatric liver cancer.
- All previous studies investigated the role of mutated b-catenin in pediatric liver cancer and in adult HCC. Our MS shows for the first time the role of the wild type b-catenin phosphorylated at Ser675 in liver cancer. In our opinion, this is highly innovative work.
Comment 2: Cells from the hepatoma cell lines, HepG2 and Huh6 cell lines are frequently used as in vitro models. I strongly suggest authors include both hepatoma cells and a normal primary cell line to support the results of a study (A comparison of p-beta-catenin-TCF4-CEGRs/ALCDs expression and pathway analysis, at the very least).
Response: This is a good suggestion; however, we think that a potential work with a normal primary cell line and interpretations of data have several problems. Every cell line in tissue culture is always a proliferating line. Our hypothesis is that b-catenin-TCF4-CEGRs/ALCDs axis is active in proliferating hepatocytes. Normal hepatocytes are quiescent cells and, if they proliferate, they display some characteristics of liver cancer (although they are not cancer cells yet). Therefore, we think that the proposed design of the study in a normal primary cell line is questionable, and that interpretation of data will be complicated. Since our manuscript presents data from three biological systems (HBL patients, PDX, and two hepatoblastoma lines), we believe that our data in HepG2 and Huh6 cells are strongly supported by the data obtained in the studies of specimens from HBL patients and by studies of PDXs.
Comment 3: The immunofluorescence staining, which the authors used to evaluate the change in b-Catenin expression in Hepatoma cells, should also be quantified.
Response. We have addressed this comment by several approaches. First, since quantitation of immunostaining is complicated and is not very accurate, we have addressed this issue by quantitation of Western Blots. Densitometric quantitation of Western blots with antibodies to ph-S675-b-catenin determined 20-25 fold elevation of ph-S675-b-catenin in tumor sections compared to non-tumor adjacent regions. This data are included in the manuscript in Fig 1C, bar graph. Second, in cultured HepG2 and Huh6 cells, we calculated numbers of ph-S675-b-catenin positive cells treated with DMSO and with PRI-724. These calculations revealed a dramatic reduction of ph-S675-positive mitotic cells treated with PRI-724. The summary of these calculations is shown as bar graphs in Figure 5G.
Comment 4: The quality images of immunofluorescence in Figures 3D and 6C need to be used.
Response: The quality of images was improved.
Comment 5: In Figure 3 D, the authors also need to compare GPC3 expression in the normal vs tumor areas.
Response: The expression of GPC3 in tumor sections was compared to its expression in non-tumor, adjacent (background) regions obtained from the same patients. Three approaches were used and showed a dramatic elevation of GPC3. These data are presented in Figs 3D and E (immunostaining), in Fig 3C (QRT-PCR, mRNA levels) and in Fig 3F (Western blot, protein levels). As one can see, these three approaches showed much higher levels of GPC in tumor sections compared to levels of GPC3 in non-tumor (background) regions.
Reviewer 2 Report
In this work, Gulati and colleagues show the relevance of the phosphorylation of b catenin in Ser675 in the development of hepatoblastoma through GECRs/ALCDs activation.
In a recent study, the authors demonstrated the activation of this pathway in fibrolamellar hepatocellular carcinoma.
In general terms, the article is well structured and written. The authors' experience in the study of the involvement of b-catenin in liver cancer is reflected in the current work.
They demonstrate in a clear way, the phosphorylation of b catenin specifically in Ser675 in HBL and how it is activated during mitosis.
The authors demonstrate via coIP how b-catenin complexes with TCF4-p300 and how it is essential for the induction of genes containing GECRs/ALCDs, using specific inhibitors.
From my point of view the experimental part is clear, I will only make minor comments:
- In material and methods the authors abuse the use of "as described in the previous work...", a brief description can be made while referring to a previous article.
-Authors should avoid the use of contractions, lines 66, 207, 440, 443, 448, 456, 484…
- Since they describe a pathway in which complexes between b-catenin, TCF4, p300 are formed to activate CEGR/ALCD, the authors could include a graphical summary to facilitate visualization of the pathway and complexes.
Author Response
We thank that the reviewer for high evaluation of our work and for useful suggestions. Our responses to the minor comments are shown below.
Comment 1. In material and methods the authors abuse the use of "as described in the previous work...", a brief description can be made while referring to a previous article.
Response: We have included brief descriptions for each section of the methods.
Comment 2. Authors should avoid the use of contractions, lines 66, 207, 440, 443, 448, 456, 484…
Response: We have corrected the text.
Comment 3. Since they describe a pathway in which complexes between b-catenin, TCF4, p300 are formed to activate CEGR/ALCD, the authors could include a graphical summary to facilitate visualization of the pathway and complexes.
Response: To address this issue, we have included b-catenin-TCF4-p300 complexes in our graphical abstract. In addition, figure 7E shows another example of a visualized summary of the role of protein-protein complexes in regulation of oncogenes via CEGRs/ALCDs regions in aggressive HBL.
Round 2
Reviewer 1 Report
I am satisfied with all corrections and responses from the authors.